



# Numerical Experiments of Cloud Seeding for Mitigating Localization of Heavy Rainfall: A Case Study of Mesoscale Convective System in Japan

Yusuke Hiraga [1*], Jacqueline Muthoni Mbugua [1], Shunji Kotsuki [2,3], Yoshiharu Suzuki [4], Shu-Hua Chen [5], Atsushi Hamada [6,2], Kazuaki Yasunaga [6], Takuya Funatomi [7]

[1] Department of Civil and Environmental Engineering, Tohoku University, Sendai, Japan
[2] Center for Environmental Remote Sensing, Chiba University, Chiba, Japan
[3] Institute for Advanced Academic Research, Chiba University, Chiba, Japan
[4] Department of Civil and Environmental Engineering, Hosei University, Tokyo, Japan
[5] Department of Land, Air and Water Resources, University of California, Davis, CA, USA
[6] Faculty of Sustainable Design, University of Toyama, Toyama, Japan
[7] Academic Center for Computing and Media Studies, Kyoto University, Kyoto, Japan

*Correspondence to*: Yusuke Hiraga (yusuke.hiraga.c3@tohoku.ac.jp)

**Abstract.** This study investigated the potential of cloud seeding to mitigate extreme rainfall localization (i.e., overseeding) associated with mesoscale convective systems in Japan. Using a numerical weather prediction model, we conducted cloud seeding experiments by artificially increasing ice nuclei concentrations in a double-moment microphysics scheme for the heavy rainfall event in Hiroshima Prefecture, Japan, in August 2014. We examined the sensitivity of rainfall changes to altitudes and areas of the seeding. The results showed that seeding in the mid–upper troposphere (7.2–8.6 km), where air temperature ranged from −22°C to −12°C, resulted in the most pronounced changes in rainfall amount. At these levels, high supercooled cloud water content and strong updrafts favored heterogeneous freezing, resulting in a depletion of moisture and suppression of graupel growth. The cloud seeding led to reduced rainfall within the heavy rainfall region and increased rainfall downstream, demonstrating the hypothesized dispersal mechanism of "overseeding". Expanding the seeding to cover the upstream region of the heavy rainfall area had a greater impact than increasing vertical thickness of the seeding. The most effective seeding configuration (24 km × 24 km area at 7.2 km) achieved an 11.5% decrease in area-averaged 3-hr accumulated rainfall and a 32% decrease as the maximum reduction in 3-hr accumulated rainfall over the heavy rainfall region. Future work should consider more realistic representations of seeding substance (i.e., transport, dispersion, and interactions) and explore a wider range of rainfall events to generalize the applicability of this approach.

## 1 Introduction

In recent years, frequency and intensity of extreme rainfall events have been increasing, leading to devastating hydrometeorological disasters worldwide (Fischer and Knutti, 2016; Papalexiou and Montanari, 2019). These trends are projected to worsen due to climate change, which is expected to further enhance the frequency and intensity of extreme rainfall





(Trenberth, 2011; Pfahl et al., 2017; Tabari, 2020). As such, developing effective mitigation strategies for extreme rainfall-induced disasters has become one of the most critical issues in the field of hydrometeorology/hydrology. Traditional hydrological measures to mitigate heavy rainfall disasters include the construction of levees and dams to improve flood safety
levels, as well as the implementation of land-use regulations and evacuation planning to reduce disaster risks (Kreibich et al., 2015). Beyond these conventional strategies, there have also been attempts to directly modify the heavy rainfall producing systems through weather modification techniques.

Weather modification techniques have historically been developed primarily for rainfall enhancement applications, aiming to augment rainfall mainly in arid and semi-arid regions (Changnon & Towery, 1990; Bruintjes, 1999; Silverman, 2010;
Murakami, 2015; Dong et al., 2021). On the other hand, weather modification studies and practices have also been conducted with the aim of weakening the disastrous weather phenomena, mainly for hurricanes (Alamaro et al., 2006; Klima et al., 2012). The most well-known historical attempts in this context were Project Cirrus (1947 – 1952) and Project Stormfury (1962–1983) run by the US government (Abe et al., 2025). These projects aimed to weaken hurricanes by dispersing precipitation through cloud seeding, which often involves the deliberate introduction of hygroscopic or ice-nucleating agents, such as silver iodide
or dry ice, into cloud systems to modify cloud dynamics and precipitation patterns. However, the effectiveness of the cloud seeding was difficult to be validated scientifically, ultimately leading to discontinuation of the project (Willoughby et al., 1985).

Recent advancements in numerical modeling, computational power, and meteorological data availability have reinvigorated interest in research on extreme rainfall mitigation. In Japan, the government has initiated the national project named "Moonshot Research and Development Program," which aims to develop feasible technologies and actions capable of suppressing extreme
rainfall intensity by 2050. This initiative has spurred active research efforts, particularly those targeting a type of mesoscale convective systems (MCSs), called "Senjo-Kousuitai", which are frequently responsible for severe localized rainfall events in Japan (Kato, 2020; Hirockawa et al., 2020). Among emerging strategies, cloud seeding with excessive amounts of seeding substances, known as "overseeding", has gained attention as a potential approach to mitigate the severity of extreme rainfall events.

The concept of overseeding was succinctly outlined in the textbooks of Mason (1971) and Rogers and Yau (1989) and nicely summarized in Durant et al. (2008), describing a scenario in which introducing an excessive quantity of ice nuclei leads to the formation of a large number of small ice crystals in convective clouds containing supercooled water droplets. Such seeding practice is categorized as glaciogenic seeding (Hashimoto et al., 2015). Under such conditions, the competition for available moisture within the cloud becomes intense, inhibiting the growth of individual ice crystals to sizes sufficient for precipitation.
When the concentration of artificially generated ice crystals significantly exceeds natural levels, the rapid increase in the number of simultaneously growing precipitation particles can result in a substantial reduction in their growth rates due to moisture depletion. Furthermore, the freezing of supercooled water releases latent heat, which strengthens the updraft and thereby reduces the sedimentation velocity of precipitation particles. Consequently, these processes would lead to a decrease in the size and sedimentation velocity of precipitation particles at the location of the overseeding, which may temporarily
suppress precipitation from the seeded cloud layer. Precipitation particles with reduced growth rates are likely advected



downstream by upper-level wind and eventually fall as precipitation in the downstream region. Thus, the overseeding has potential to mitigate the localization of intense precipitation and facilitate its dispersion over a wider area, which may be effective in reducing disaster risk.

The aforementioned concept of the overseeding is well-described in the recent studies (Koloskov et al., 2010; Murakami, 2015; Korneev et al., 2022; Abshaev et al., 2022). To date, previous overseeding experiments have not necessarily been conducted with the primary objective of mitigating extreme rainfall. If this concept is extended to extreme rainfall events, overseeding could potentially serve as a mitigation strategy for heavy rainfall disasters. So far, the application of overseeding and related numerical experiments have generally been limited. Recently, a series of the numerical experiments of overseeding in heavy rainfall events over Japan have demonstrated its potential for altering total precipitation and peak rainfall intensity (Suzuki et al., 2012; Yokoyama et al., 2015; Nozaki et al., 2024). Although such experiments have yielded promising results, our understanding of the mechanisms of overseeding and the effective conditions for overseeding remains limited. Further studies are required to assess the effectiveness of cloud overseeding in different conditions, in order to better understand the necessary conditions of overseeding for effectively mitigating heavy rainfall and its feasibility.

Based on the discussion above, this study assesses the effectiveness of cloud overseeding, by numerical experiments in which the cloud seeding are conducted at different conditions within a target convective system. The findings of this study provide a valuable foundation for understanding the potential of cloud overseeding and the optimal conditions for effectively mitigating heavy rainfall in convective systems. Section 2 describes the target convective rainfall event, numerical model, data, and the experimental settings for cloud seeding. Section 3 presents the numerical results. Section 4 discusses the results of the numerical experiment and its feasibility for real-world application. Section 5 provides summary, limitations, and future research directions.

## 2 Materials and Methods

### 2.1 Target convective heavy rainfall event

This study focuses on the convective heavy rainfall event that occurred around Hiroshima City, Japan, in August 2014. The event produced extreme precipitation, exceeding 100 mm/h and 240 mm over a three-hour period, leading to catastrophic flooding and landslides that resulted in 75 fatalities and the destruction of 330 houses (Hirota et al., 2016; Oizumi et al., 2020). This event was characterized by an intense line-shaped rainband, approximately 100 km in length and 20–30 km in width (Figure 1c). Such quasi-stationary line-shaped rainbands, known as Senjo-Kousuitai in Japanese due to their distinctive shape, have garnered significant attention as they contribute to severe flood disasters nearly every year in Japan (Kato, 2020). The back-building type of strong multi-cell convective systems, which initiated in the mountainous area in southwestern Hiroshima Bay (Figure 1), resulted in such line-shaped rainfall band (Kato, 2020; Oizumi, 2020). The warm and moist southern inflow through the Bungo channel, where the convective instability was quite high, with the orographic uplift, favored the initiation of strong convections (Kato, 2020; Oizumi, 2020). An unstable atmospheric stratification associated with the cold core of a



cutoff low in the upper troposphere and abundant free-tropospheric moisture played an important role in causing deep convection and precipitation (Hirota et al., 2016). There have been extensive studies on the mechanisms and natural disasters

associated with this rainfall event in Hiroshima in 2014 because this event represents the typical characteristics of disastrous convective rainfall events in Japan (Wang et al., 2015; Hibino et al., 2018). Thus, this event serves as a valuable case study for assessing the impacts of cloud overseeding on disastrous convective rainfall events in the region. The fact that such convective heavy rainfall events are projected to intensify and increase under future climate (Kawase et al., 2023; Hiraga et al., 2025) further highlights the importance of understanding the effectiveness of cloud seeding for such systems as a counter measure.

Our analysis mainly focused on the 3-hour rainfall amount from 16:00 to 19:00 UTC on August 19, 2014, which covers the intense rainfall during the target event. We defined the "heavy rainfall region" as the area where the 3-hr rainfall accumulation exceeded 100 mm. This threshold is commonly used to assess the risk of landslide disasters in Japan (MLIT, 2007).

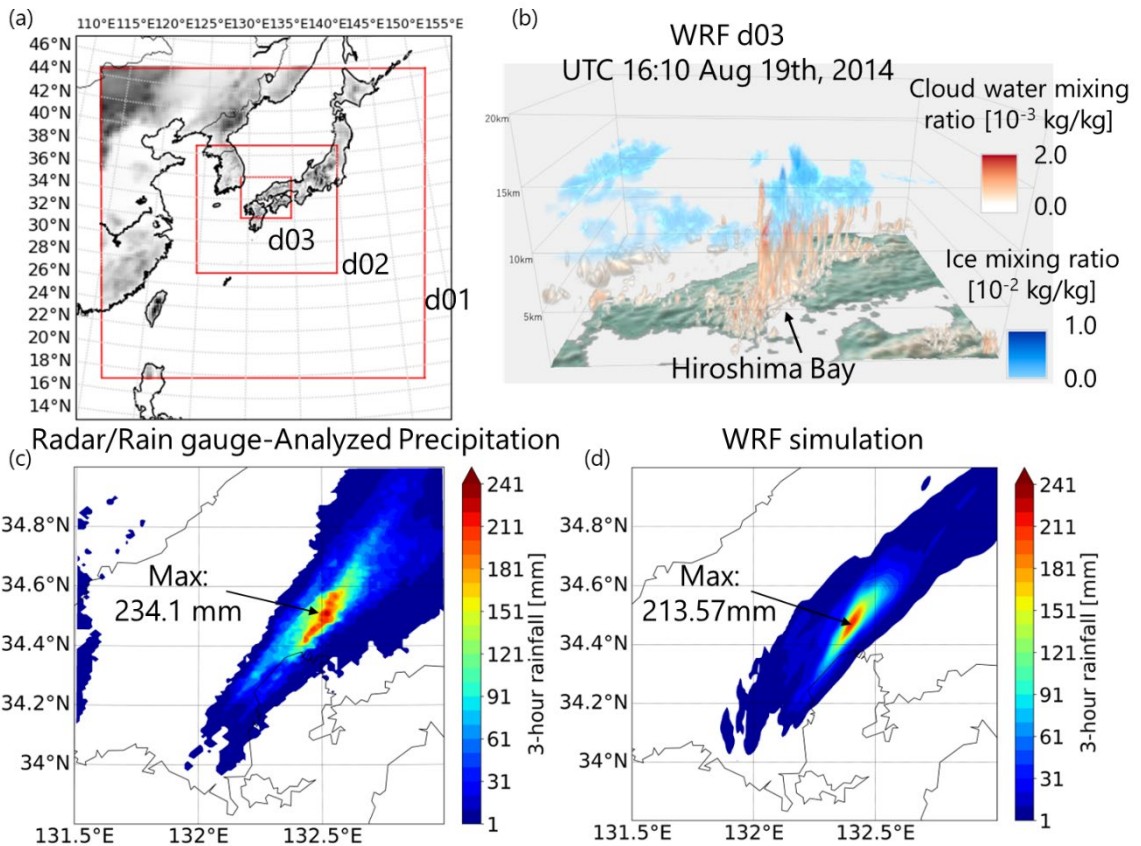

**Figure 1 (a) WRF model domains: d01 (25 km), d02 (5 km), and d03 (1 km); (b) Three-dimensional distribution of**
**WRF-simulated hydrometeors at 16:10 UTC on August 19, 2014; (c) Radar/Rain gauge-Analyzed Precipitation-based**
**3-hr accumulated rainfall from 16:00 to 19:00 UTC on August 19, 2014; (d) WRF-simulated 3-hr accumulated rainfall**
**for the same period.**



## 2.2 Model and Data

This study used the Advanced Research version of the Weather Research and Forecasting model (WRF), version 4.1.2, for numerical experiments. WRF is a fully compressible and non-hydrostatic model that uses terrain-following hydrostatic-pressure vertical coordinates and Arakawa C-grid staggering spatial discretization for atmospheric variables (Skamarock et al., 2019). WRF has been widely used to simulate quasi-stationary line-shaped rainfall (Kawano and Kawamura, 2020; Nakanishi, 2024; Hiraga and Tahara, 2025; Tahara et al., 2025). The WRF dynamically downscales a given meteorological input over a

configured nested domain while solving nonlinear governing equations and parameterizing subgrid-scale processes such as microphysics, boundary layer eddies, and cumulus clouds. In this study, the configurations of the model domains and physics parameterizations basically followed Kita et al. (2016) and Oizumi et al. (2020), who successfully simulated the same rainfall event. A key distinction from these studies is the use of Morrison 2-moment cloud microphysics scheme to employ the cloud seeding experiment. The Morrison double-moment microphysics scheme (Morrison et al., 2005, 2009; Morrison and Milbrandt,

2015) is a bulk microphysics parameterization that predicts both the mass and number concentration of hydrometeor species, allowing for a physically consistent representation of cloud microphysical processes. This scheme explicitly accounts for interactions with cloud condensation nuclei (CCN) and ice nuclei (IN), and includes prognostic variables for cloud droplets, rain, ice, snow, and graupel, enabling a detailed simulation of phase changes, including freezing, condensation, evaporation, deposition, and riming. Such capability makes it particularly useful for studies investigating the microphysical impacts of

aerosol-cloud interactions, cloud seeding, and extreme precipitation events. The scheme has been widely implemented in cloud-resolving models due to its balance between computational efficiency and physical realism (Mohan et al., 2018; Huang et al., 2020). Three computational domains with grid resolutions of 25 km, 5 km and 1 km were configured with a two-way nesting approach (Figure 1a). The third domain (d03), with a horizontal resolution of 1 km, encompasses the Chugoku and Shikoku regions of Japan including Hiroshima prefecture. Previous studies suggested that the spatial resolution of 2 km or

higher is generally ideal for simulating quasi-stationary band-shaped rainfall system (Kato et al., 2020), which is satisfied in our domain configuration. Additional model settings used in this study are summarized in Table 1.

**Table 1 WRF physics parameterization settings used in the study**

| Physics parameterizations | Schemes |
| --- | --- |
| Cumulus convection (only d01) | Kain–Fritsch (Kain, 2004) |
| Cloud microphysics | Morrison 2-momnet (Morrison et al., 2009) |
| Shortwave radiation | RRTMG (Iacono et al., 2008) |
| Longwave radiation | RRTMG (Iacono et al., 2008) |
| Planetary Boundary Layer | MYNN 2.5 (Nakanishi and Niino., 2006; 2009) |



| Surface Layer | Revised MM5 (Jimenez et al., 2012) |
|---|---|
| Land surface processes | Noah-MP Land Surface Model (Niu et al., 2011; Yang et al., 2011) |

The initial and boundary conditions for the WRF simulations were from NCEP Global Data Assimilation System (GDAS) FNL operational global analysis. The NCEP GDAS FNL is available at 6-hour intervals with a 0.25° horizontal resolution and 34 vertical levels. We used the Radar/Raingauge-Analyzed Precipitation observation data (RA data) to verify the WRF-simulated rainfall. The RA data has been widely used as ground-truth rainfall data to assess the accuracy of simulated precipitation owing to its high accuracy and spatial/temporal resolution (hourly and 1 km) (e.g., Minamiguchi et al., 2018;

Nakanishi, 2024; Hiraga and Tahara, 2025).

## 2.3 Experimental settings

The target heavy rainfall event was first simulated using the configured WRF model to ensure its credibility of reproducing the observed rainfall band (hereafter referred to as the CTL run). The simulation successfully captured the convective heavy

rainfall, as confirmed by comparison with RA-based observations (Figures 1c versus 1d). The back-building structure of deep convection was also well represented (Figure 1b), supporting the use of this simulation as the baseline for the overseeding experiments. We adopted the spin-up time of 16 hours for all the computations (i.e., the model integration started at 00:00 UTC on August 19, 2014).

Next, we performed cloud overseeding experiments by modifying the CTL run based on the concept of overseeding to

investigate the potential of the overseeding for mitigating the heavy rainfall. We represented cloud overseeding in the numerical simulation by artificially increasing the number of ice nucleus concentration, following Suzuki et al. (2012), Yokoyama et al. (2015), and Nozaki et al. (2024). It is well established that a large number of ice nuclei can be artificially generated using silver iodide (AgI) or dry ice (Fukuta et al., 1971). The method used the unitless multiplier in Meyer's formula shown in Eq. (1) (Meyers et al., 1992), to represent such a large increase in ice nuclei. In WRF v4.1.2, the Meyer's formula is

implemented within the Morrison 2-moment cloud microphysics scheme.

$$n_c = \exp\{-2.80 + 0.262 \times (273.15 - T)\} \times \beta \qquad (1)$$

where $n_c$ is the number of ice nucleus concentration per kilogram, $T$ is air temperature, and $\beta$ is a unit less multiplier. In the WRF model, the Meyer's formula triggers the freezing of cloud droplets when the following conditions are met: a cloud water mixing ratio greater than $10^{-14}$ kg kg$^{-1}$ and an air temperature below –4°C (i.e., deposition freezing). Following previous studies (Suzuki et al., 2012; Yokoyama et al., 2015; Nozaki et al., 2024), a large unitless multiplier ($\beta$=10$^9$) was adopted to represent

the overseeding condition. The number of ice nuclei generated by AgI can reach an exceedingly large value, ranging from



$10^{10} \sim 10^{16}$ per gram of AgI (Murakami, 2015). Dry ice is known to produce an astronomically large number of ice crystals through the deposition process ($10^{13}$ per gram of dry ice) (Fukuta et al., 1971; Murakami, 2015). The generation of such an extreme concentration of ice crystals releases latent heat, potentially invigorating updrafts in the mixed-phase clouds and transporting large quantities of ice crystals upward. As such, moisture depletion and inhibited droplet growth are expected near
the cloud top.

     The overseeding experiments followed the steps outlined below:

     (1) We first conducted the overseeding experiment at a location aligned with the long axis of the line-shaped rainfall band, where deep convection occurred in the CTL simulation (Figure 2a). This location was determined based on a detailed examination of vertical cross-sections of hydrometeors within the convective system. At this site with 6 km × 6 km area shown
in Figure 2a, overseeding was performed across eight distinct vertical layers to investigate the sensitivity of rainfall changes to the seeding height. The selected layers satisfied the conditions under which Meyer's formula is activated, aligning with the concept of overseeding into supercooled clouds: a cloud water mixing ratio greater than $10^{-14}$ kg kg$^{-1}$ and an air temperature below –4°C. Each vertical layer corresponds approximately to 5.7 km, 6.4 km, 7.2 km, 7.9 km, 8.6 km, 9.3 km, 10 km, and 10.7 km from the ground surface (corresponding indices of model vertical layers are 20, 22, 24, 26, 28, 30, 32, and 34 from
the surface as 0, respectively), covering the different heights of supercooled clouds to the top. As shown in Figure 2, the designated seeding locations cover the deep convection during the early back-building development phase (Figures 2b), back-building sustained to ending phase (Figure 2c), and late rainfall localization phase (Figures 2d).

     (2) Subsequently, we aimed to enhance the effectiveness of overseeding in order to induce further modifications in rainfall. To this end, the seeding area was expanded by broadening the horizontal extent upstream. The intention was to perform
overseeding on the early-phase convection to suppress the growth of precipitation particles there, which could influence the subsequent development of the convective system, given that the target event was of the back-building type. The expanded areas correspond to the total of 8 cases: 9 km × 9 km, 12 km × 12 km, 15 km × 15 km, 18 km × 18 km, 21 km × 21 km, 24 km × 24 km, 27 km × 27 km, and 30 km × 30 km. During this horizontal area expansion experiment, the vertical seeding height remained fixed at 7.2 km, which was found to be the most effective layer in step (1).
(3) Additionally, we conducted experiments in which the vertical thickness of the seeded layer was increased, considering that the effectiveness of seeding depends on the vertical level of injection. Specifically, the vertical layer thickness was set to 1 layer (at approximately 7.2 km), 3 layers (at approximately 6.8–7.6 km), and 5 layers (at approximately 6.4–8.0 km), respectively. In this thickening experiment, the horizontal seeding location was kept the same as in the original configuration shown in Figure 2.
Table 2 summarizes the experimental flow. Upon identifying the effective overseeding conditions, their feasibility was discussed in terms of the required amount of seeding materials.





**Table 2 Experimental flow**

|  | Aims and objectives | Experimental settings |
| --- | --- | --- |
| Step (1) | To evaluate whether the implemented seeding methodology produces the hypothesized overseeding effects<br><br>To investigate the sensitivity of rainfall responses to changes in seeding altitude | Seeding with 6 km × 6 km area at the folloing vertical heights (model vertical layer#): 5.7 km (layer 20), 6.4 km (layer 22), 7.2 km (layer 24), 7.9 km (layer 26), 8.6 km (layer 28), 9.3 km (layer 30), 10 km (layer 32), 10.7 km (layer 34) |
| Step (2) | To enhance the effectiveness of overseeding by expanding the horizontal extent of the seeding area upstream<br><br>*Supplementary Material examines the effectiveness of dividing the seeding area into upstream and downstream regions. | Seeding at 7.2 km over the folloing horizontal extents:<br>9 km × 9 km, 12 km × 12 km, 15 km × 15 km, 18 km × 18 km, 21 km × 21 km, 24 km × 24 km, 27 km × 27 km, and 30 km × 30 km |
| Step (3) | To enhance the effectiveness of overseeding by increasing the vertical thickness of the seeding layers | Seeding with 6 km × 6 km area at the folloing vertical layers:<br>1 layer (at approximately 7.2 km), 3 layers (at approximately 6.8–7.6 km), and 5 layers (at approximately 6.4–8.0 km) |



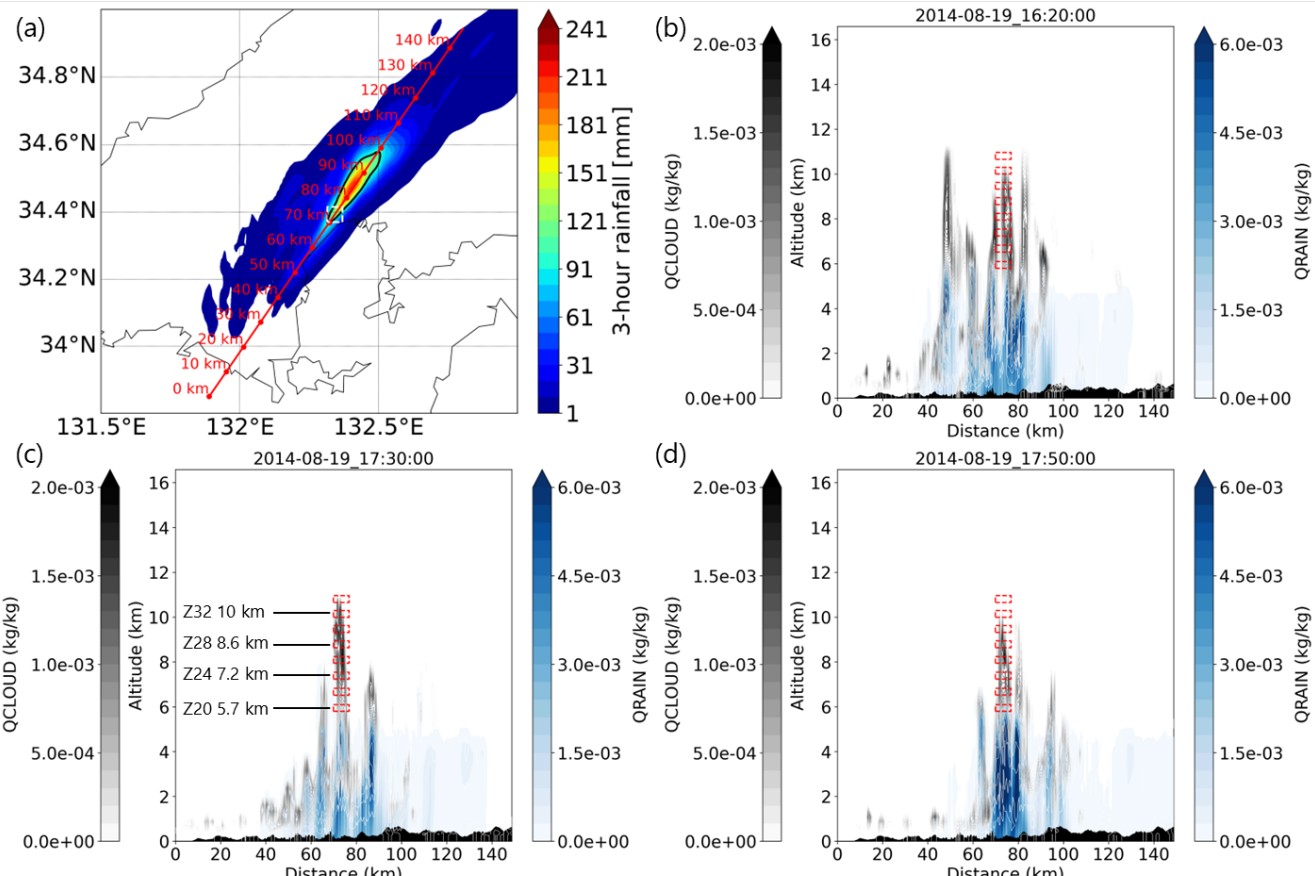

**Figure 2.** (a) WRF-simulated 3-hr accumulated rainfall with the path for vertical cross sections (red line and points), horizontal extent of cloud seeding (white dashed square), and heavy rainfall region (>100 mm in 3 hours; black outline); (b–d) vertical cross sections of cloud water mixing ratio (QCLOUD; gray contours) and rainwater mixing ratio (QRAIN; blue contours) at different times, with cloud seeding locations indicated at each specified altitude (red squares). In (c), the numbers after Z indicate the model layers and heights where cloud seeding occurs.

## 3 Results

### 3.1 Overseeding experiments with different vertical layers

Our experimental design, which increased the concentration of ice nuclei in the microphysics scheme, successfully represented the anticipated processes associated with overseeding. Figure 3 illustrates the impact of seeding at the 7.2 km level on hydrometeor profiles. At and above the vertical level where seeding was performed, the ice mixing ratio significantly increased, while the cloud water mixing ratio decreased, implying enhanced heterogeneous freezing (Figures 3a and 3b). The enhanced heterogeneous freezing led to the release of latent heat and an associated increase in updraft intensity, as shown in

 

the arrows in each panel, which in turn contributed to the vertical transport of ice nuclei to higher atmospheric layers. This
process led to competitive moisture uptake and subsequent depletion at and above the seeding layer (Figure 3b). The depletion
of cloud water and the presence of numerous ice crystals generally inhibit the riming process, thereby reducing the formation
and growth of larger hydrometeors such as graupel (Figure 3c). An increase in ice mixing ratio and decreases in both cloud
water and graupel mixing ratios at the seeding location and its near downstream were consistently observed throughout the
seeding period (Supplementary Material; Figure S1). The changes initiated at the seeding location were eventually propagated
downstream. By 17:50 UTC (Figures 3d and 3e), localized reductions in cloud water and graupel mixing ratios near the seeding
area were evident, with corresponding increases observed approximately 100 km downstream. This spatial pattern supports
the hypothesis that seeding-induced moisture depletion inhibited hydrometeor development near the source, promoting
downstream transport by the prevailing winds and subsequent development of particles. As a result, the rainwater mixing ratio
significantly decreased in the heavy rainfall region in the CTL run (~80 km) and increased downstream (~100–120 km),
facilitating the dispersion of localized heavy rainfall over a broader area (Figure 3f). Overall, our analysis above clearly
demonstrates that the experiment successfully reproduced the anticipated processes and outcomes associated with overseeding.

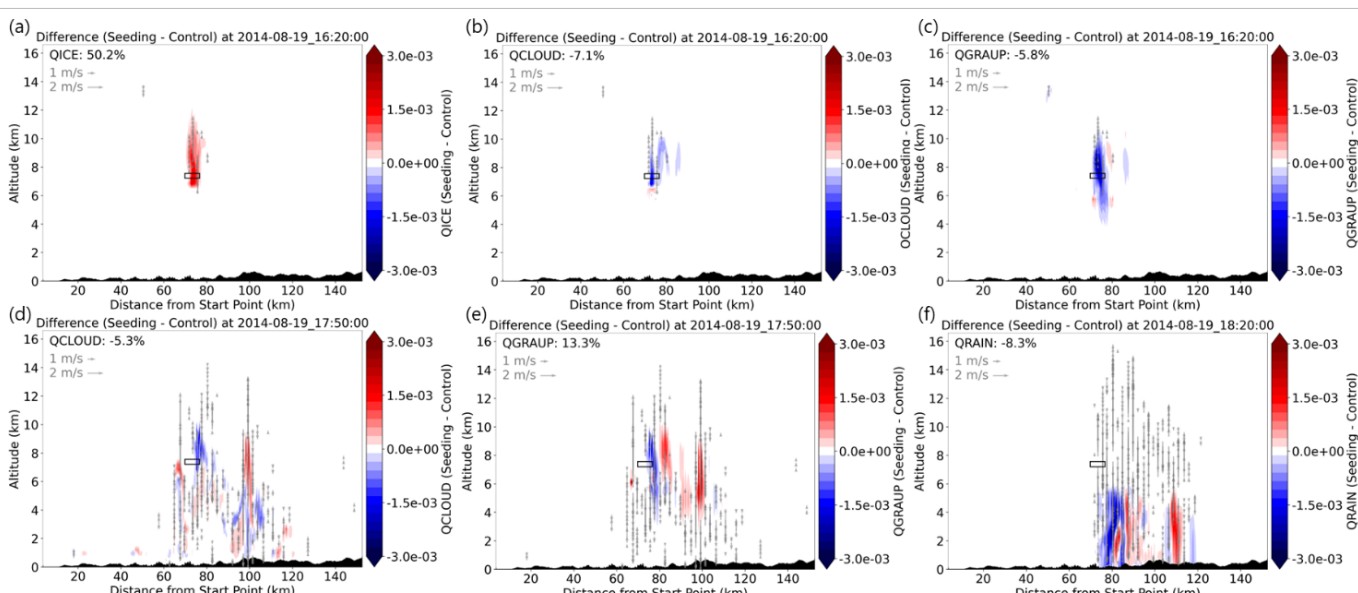

**Figure 3. (a-c) Differences between the seeding experiment (6 km × 6 km at 7.2 km altitude) and the control (CTL) run in ice mixing ratio (QICE) [kg kg⁻¹], cloud water mixing ratio (QCLOUD) [kg kg⁻¹], and graupel mixing ratio**
**(QGRAUP) [kg kg⁻¹] at 16:20 UTC on August 19, 2014; (d, e) differences in QCLOUD and QGRAUP at 17:50 UTC on August 19, 2014; (f) difference in rainwater mixing ratio (QRAIN) [kg kg⁻¹] at 18:20 UTC on August 19, 2014. The percentage change shown in each panel represents the cross-sectional average change in hydrometeors in the displayed extent.**





Figure 4 illustrates the changes in 3-hour accumulated rainfall resulting from seeding at different vertical layers (5.7 km, 6.4 km, 7.2 km, and 10 km). The results of the seeding at the other vertical layers are shown in Supplementary Material (Figure S2). Although differences among the seeding heights were not significant, seeding near the middle of the tested layers (e.g., at 7.2 km; Figure 4b) tended to reduce rainfall more and over a broader area. In Figure 4b, the pattern of rainfall change is characterized by a decrease in the heavy rainfall region (i.e., > 100 mm in 3-hr accumulation in the CTL run) and an increase

downstream, which is consistent with our hypothesis, despite the relatively small reduction in rainfall within the heavy rainfall region, indicated by the black perimeter (an average change of –3.3 mm; –2.3%). The effects of overseeding accumulated over time, highlighting that the changes were not achieved instantaneously (Supplementary Material; Figure S3). The change in rainfall became more pronounced after 17:00 UTC.

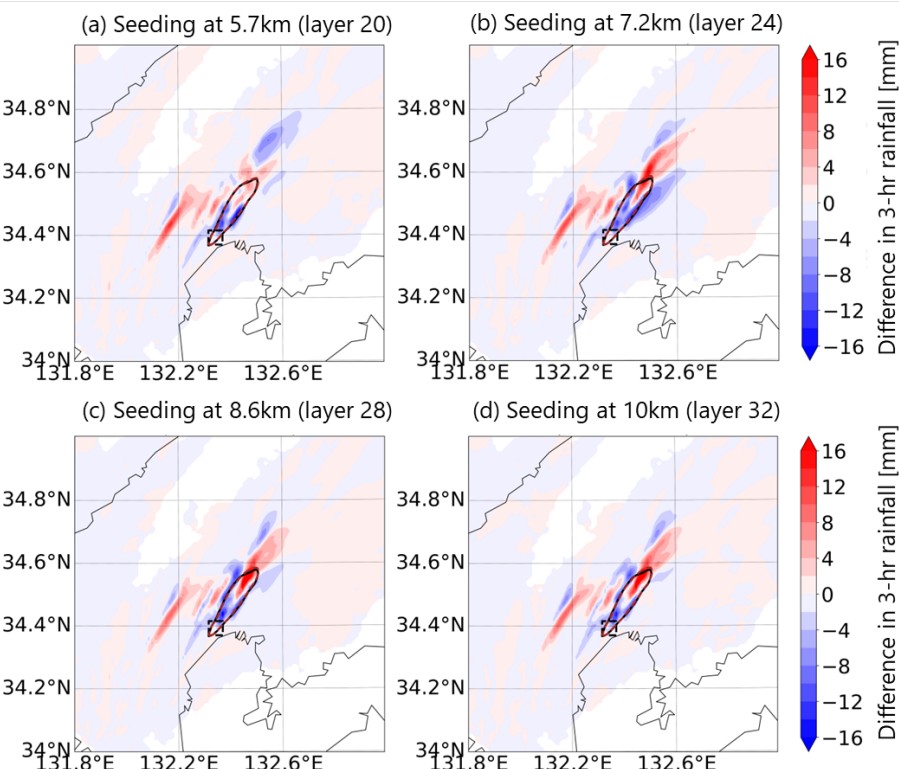

**Figure 4. (a–c) Differences in 3-hr accumulated rainfall between the seeding experiments (6 km × 6 km at each height) and the control (CTL) run from 16:00 to 19:00 UTC on August 19, 2014.**

We investigate the physical mechanisms underlying the pronounced seeding effect observed at the 7.2 km level compared to other altitudes. As shown in Figure 5, early changes in ice mixing ratio varied depending on the seeding height. Seeding at

a relatively low altitude (5.7 km) did not produce a substantial increase in ice mixing ratio at the seeding location, indicating limited seeding effectiveness. In contrast, seeding at middle altitudes (7.2 km and 8.6 km) led to a notable enhancement in both ice mixing ratio and updraft strength, reflecting a strong seeding effect. At a further higher seeding altitude (10 km), the





impact was weakened, with smaller changes in ice mixing ratio observed. To further examine the mechanisms underlying the

height-dependent differences in seeding impacts, we investigated the background atmospheric conditions. Figure 6 presents

vertical cross-sections of air temperature, cloud water mixing ratio, and vertical velocity. The seeded altitudes where large

increases in ice water mixing ratio were observed (7.2 km–8.6 km), exhibited air temperatures ranging from −12°C to −22°C

during the seeding period. The lowest analyzed layer (5.7 km) showed temperatures between −4°C and −6°C, while the highest

layer (10 km) exhibited temperatures below −30°C. Both the cloud water mixing ratio and vertical velocity peaked near the

mid-tropospheric levels, particularly at 7.2 km, within the cumulus cloud cell, whereas values were lower in the upper and

lower layers. Overall, Figure 6 suggests that seeding at the mid-tropospheric levels (around 7.2 km to 8.6 km) effectively

introduced seeding material into a region characterized by the presence of large amount of supercooled water droplets and

strong updrafts. This likely enhanced freezing processes, leading to heterogeneous ice formation not only at the seeded layer

but also in the layers above.





**Figure 5. Differences between the seeding experiment (6 km × 6 km at each altitude) and the control (CTL) run in ice mixing ratio (QICE) [kg kg⁻¹] at 16:20 UTC on August 19, 2014. The percentage change shown in each panel represents the cross-sectional average change in hydrometeors in the displayed extent.**

**Figure 6. Vertical cross-sections of air temperature (left column), cloud water mixing ratio (middle column), and vertical velocity (right column) at 16:20 UTC (upper raw), 16:50 UTC (middle raw), and 18:00 UTC (bottom raw) on August 19, 2014 in the CTL run.**

## 3.2 Experiments to enhance the overseeding impacts

The results obtained in Section 3.1 demonstrated that the employed overseeding method successfully produced the intended overseeding effects, leading to rainfall responses consistent with our hypothesis. However, the resulting impacts were relatively modest, with the best case showing an average reduction of –3.3 mm (–2.3%) within the heavy rainfall region. Therefore, the subsequent experiments were designed to amplify the effects of overseeding on rainfall.





The impact of seeding on rainfall was considerably enhanced when the seeding area was expanded by broadening the
horizontal extent upstream. Figure 7 presents the changes in 3-hr accumulated rainfall resulting from seeding over the extended
upstream areas (i.e., dashed-line boxes in Figures 7a-f). Overall, the expansion of the seeding region led to more pronounced
changes in rainfall, consistently characterized by a reduction in the heavy rainfall region and an increase in rainfall downstream.
Although the cases with expanded seeding areas further increased rainfall downstream, such as seeding over 24 km × 24 km
(Figure 7e), this increase did not lead to a notable expansion of the heavy rainfall region relevant to landslides. A comparison
of Figures 7g and 7h clearly illustrates this behavior, characterized by a reduction in localized intense rainfall and a downstream
expansion of precipitation without the development of new heavy rainfall regions.

Namely, expanding the seeding area was shown to be effective in preventing the localization of rainfall by promoting its
dispersion over downstream. Among all cases shown in Figure 7, the case of seeding over 24 km× 24 km led to the largest
reduction in area-averaged 3-hr accumulated rainfall within the heavy rainfall region (-16.3mm; -11.5%). Interestingly, further
expansion of the seeding area beyond 24 km × 24 km did not lead to additional changes in rainfall, suggesting the existence
of an optimal seeding configuration.

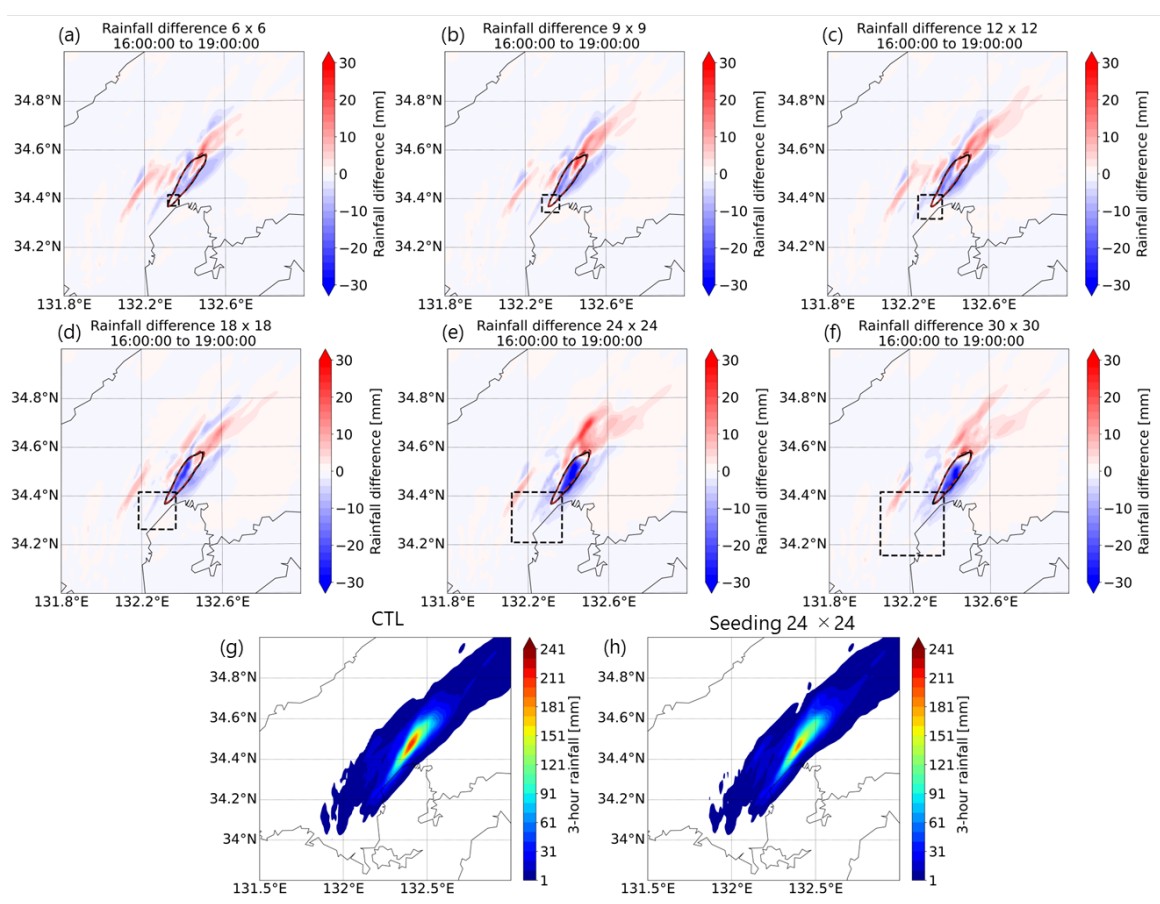



**Figure 7. (a–f) Differences in 3-hr accumulated rainfall between the seeding experiments (different spatial extent at 7.2km altitude) and the control (CTL) run from 16:00 to 19:00 UTC on August 19, 2014; (g and h) 3-hr accumulated**
**rainfall in the CTL run and seeding run (24 km × 24 km at 7.2km altitude).**

The changes in rainfall accumulation due to the seeding practices are summarized as histograms for a quantitative comparison (Figure 8). Seeding over the 24 km × 24 km area led to predominantly negative changes in 3-hr rainfall within the heavy rainfall region with the largest reduction reaching 45.4 mm (−32.0%). The seeding over 24 km × 24 km also mitigated
the increase in rainfall within the heavy rainfall region (shown in the green dashed line in Figures 8a-8c) compared to the cases of seeding over 6 km × 6 km and 12 km × 12 km, highlighting its promising result for preventing rainfall localization. However, the 24 km × 24 km seeding configuration induced greater rainfall increases further downstream (up to 23.7 mm; 310.5% in Figure 8f), compared to the 12 km × 12 km case resulting in a smaller maximum increase of 18.45 mm (241.8%; Figure 8e). In fact, seeding over the 12 km × 12 km area already resulted in a meaningful mitigation of rainfall within the heavy rainfall
region (e.g., minimum of 12.3 % reduction in the heavy rainfall region). Balancing the trade-off between rainfall suppression in heavy rainfall regions and enhancement in downstream areas is critical for the practical design and implementation of cloud overseeding strategies.



**Figure 8. (a–f)** Histograms of 3-hr accumulated rainfall differences between the seeding runs and the control (CTL) run. Panels (a, d) correspond to the 6 km × 6 km seeding at 7.2 km altitude; (b, e) to the 12 km × 12 km seeding at 7.2 km; and (c, f) to the 24 km × 24 km seeding at 7.2 km. The left column shows differences over the heavy rainfall region, while the right column shows differences over the broader region from 131.8°E to 132.8°E and 34°N to 35°N.

Our analysis of the physical mechanisms underlying the rainfall changes revealed that further expansion of the seeding area captured upstream convective cells, subsequently enhancing the overall impact of seeding. As shown in the left and middle columns of Figure 9, expanding the seeding area to 9 km × 9 km and 12 km × 12 km did not lead to noticeable enhancement





in the changes of ice and graupel mixing ratios, compared to the seeding area of 6 km × 6 km (Figure 3). In contrast, seeding
over the 24 km × 24 km area triggered overseeding conditions in further upstream region at 40-60 km, leading to enhanced
hydrometeor changes within the upstream convective cell (right column; Figures 9c and 9f). Such early changes in
hydrometeors gradually propagated downstream with vertical oscillations, as illustrated at 16:40 UTC (Figures 9g-9i).
Capturing the upstream convective cell in the 24 km × 24 km seeding case eventually resulted in a stronger change in graupel
mixing ratio by 17:50 UTC, with enhanced reduction in the heavy rainfall region (around 80 km) and increased accumulation
downstream (around 100-120 km), relative to cases that failed to influence the upstream cell (Figures 9j-9l).

**Figure 9. (a–c) Differences between the seeding experiment (9 km × 9 km, 12 km × 12 km, and 24 km × 24 km at 7.2
km height) and the control (CTL) run in ice mixing ratio (QICE) [kg kg⁻¹] at 16:20 UTC on August 19, 2014; (d-f) Same**


**as a-c but for graupel mixing ratio (QGRAUP) [kg kg⁻¹]; (g-i) Same as d-f but at 16:40 UTC on August 19, 2014; (j-l) Same as d-f at 17:50 UTC on August 19, 2014. The percentage change shown in each panel represents the cross-sectional average change in hydrometeors in the displayed extent.**

Such seeding practices influence the structural characteristics of convective cells, leading to observable modifications in their organization and intensity. Figure 10 shows the 3-D distribution of the ice number concentration and cloud water mixing ratio in the CTL (left column) and seeding (right column) runs. At earlier times following the seeding (at 16:30 UTC), a significant increase in ice number concentration (on the order of $10^6$ kg⁻¹) was observed at the seeding location (Figures 10a and 10b). Such an increase in ice number concentration corresponds to a reduction in the cumulus cloud top height—defined as the altitude at which the cloud water mixing ratio exceeds 0 kg/kg—as well as a decrease in the cloud water mixing ratio within the region of heavy precipitation. By 17:50 UTC, a significant increase in both cloud water mixing ratio and its distribution density due to seeding became evident in the downstream region (Figures 10c and 10d). Meanwhile, a reduction in cloud water mixing ratio can be observed in the original heavy rainfall region. This clearly illustrates a shift in intense cumulus cloud development from the heavy rainfall region to the downstream area.

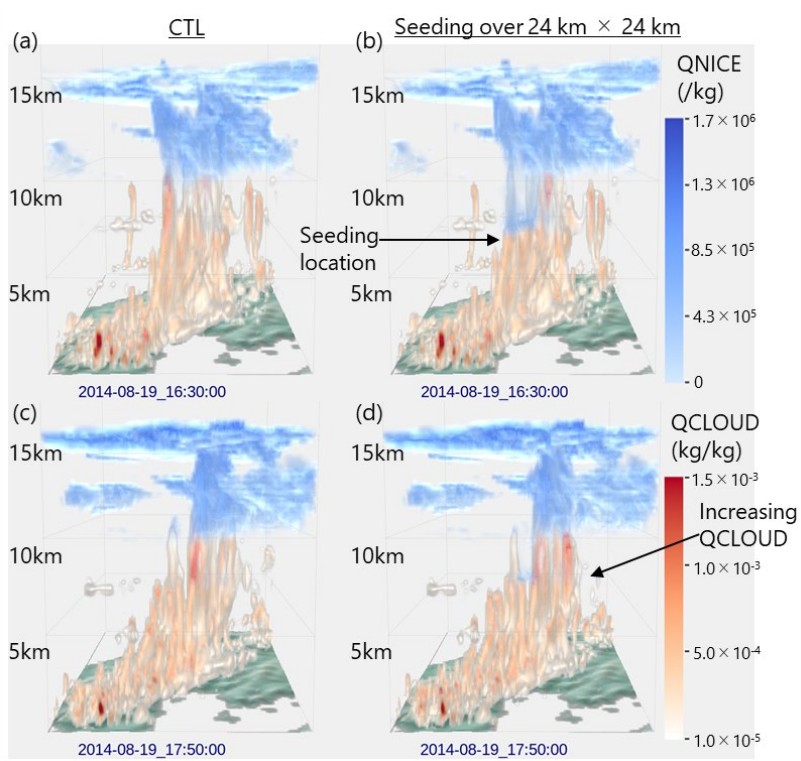

**Figure 10. (a and b) Three-dimensional distribution of WRF-simulated hydrometeors (QNICE: Ice number concentration [kg⁻¹]; QCLOUD: Cloud water mixing ratio [kg kg⁻¹]) at 16:30 UTC on August 19, 2014, for the control (CTL) run and the 24 km × 24 km seeding run; (c, d) same as (a, b) but at 17:50 UTC on August 19, 2014.**



Given that expanding the seeding area to include upstream convective cells was found to be effective, this study further explored a hypothetical scenario: seeding applied solely in the upstream region. We divided the 24 km × 24 km square shown in Figure 7 into two 12 km × 12 km subdomains, with one covering the upstream and the other covering the downstream. Here,

the upstream square was designed to encompass the upstream convective cell shown in Figure 9c. As a result, shown in in Supplementary Materials (Figures S7 and S8). seeding applied solely in the upstream or downstream region was less effective in terms of 3-hr rainfall changes compared to seeding over the full 24 km × 24 km area. This finding highlights the importance of capturing convective cells over a broader spatial extent. Moreover, increasing the vertical thickness of the seeding layers was generally found to be less effective than expanding the horizontal extent of the seeding area. This experiment is shown in

Supplementary Materials (Figures S7 and S8). Starting from the baseline case of seeding over a 6 km × 6 km area at a height of 7.2 km, we increased the vertical thickness of the seeding layer from 1 to 3 and 5 vertical layers. As a result, the differences in the average changes in rainfall accumulation over the heavy rainfall region were minimal: –3.0 mm, –2.5 mm, and –2.8 mm for vertical layer thicknesses of 1, 3, and 5 layers, respectively. This finding suggests that expanding the horizontal extent of the seeding area may be more effective than increasing the vertical thickness of the seeding layer. Further investigations

involving a wider range of seeding settings and different MCS events are necessary to generalize this result. Such experiments are left for future work.

## 4 Discussion

### 4.1 Discussion of the findings

This study designed numerical cloud overseeding experiments following previous studies (e.g., Nozaki et al., 2024), in

which the ice nucleus concentration was artificially increased in Meyer's formula within the Morrison double-moment microphysics scheme. Overall, our experiment successfully demonstrated the hypothesized seeding effects (i.e., overseeding): the introduction of a large number of ice crystals into supercooled clouds, subsequent moisture depletion and strengthening of the updraft, and the resulting temporary suppression of precipitation from the seeded cloud layer with precipitation being dispersed downstream. It should be noted that changes in convective system development due to seeding are highly non-linear

and strongly influenced by atmospheric chaos, making it difficult to simply interpret the propagation and extent of seeding impacts. Thus, it is important to note that our findings from the seeding experiments may be significantly influenced by atmospheric chaotic behavior, including various factors beyond the direct and intuitive impacts of seeding.

Our findings indicate that the effectiveness of seeding varies with the vertical layer at which it is applied. This sensitivity to seeding altitude is consistent with previous studies (Onaka and Suzuki, 2014; Yokoyama et al., 2015; Sano et al., 2024). A

pronounced reduction in rainfall over the heavy rainfall region was observed when seeding was conducted near altitudes of 7.2–8.6 km, where air temperatures ranged from −12°C to −22°C, and conditions were characterized by high cloud water mixing ratios (i.e., supercooled water) and strong updrafts. This temperature range corresponds to an atmospheric layer often



deficient in natural ice-forming nuclei and has been identified as an effective target region for cloud seeding (Grant and Elliott, 1974). Our finding is generally in agreement with the results of Sano et al. (2024), who reported a greater impact of seeding

in the lower layer (4.61–8.19 km) compared to the higher layer (8.19–11.44 km). While our findings are generally consistent with previous studies in terms of hydrometeor behavior and the reduction of localized heavy rainfall, those studies did not explicitly address the analysis of downstream dispersion of rainfall and its mechanism. This study clearly demonstrated the hypothesized overseeding mechanism, whereby rainfall in the original heavy rainfall region is dispersed downstream in a quantitative way. Consideration of such downstream rainfall increases is critical for designing seeding experiments in practical

applications.

Our analysis revealed that expanding the horizontal extent of the seeding area in the upstream direction induced more pronounced changes in rainfall accumulation than increasing the vertical thickness of the seeding layer. The case of seeding over 24 km × 24 km resulted in the largest reduction in 3-hr accumulated rainfall (-45.4 mm; -32.0%) as well as the area-averaged 3-hr accumulated rainfall (-16.3mm; -11.5%) within the heavy rainfall region. The main difference between seeding

over the expanded area and seeding over the smaller area was the appearance of seeding effects in upstream convective cells, which ultimately led to more pronounced changes in the distribution of hydrometeors along the target MCS. Our best results on rainfall reduction are comparable to, or even exceed, those reported in recent numerical modeling studies aimed at mitigating heavy rainfall through seeding (Nozaki et al., 2024; Sano et al., 2024), suggesting the high sensitivity of the targeted MCS to seeding. Further investigations involving a wider range of seeding configurations and different MCS events are

necessary to generalize the observed rainfall changes resulting from the overseeding experiment, underscoring the importance of continued research in this field.

## 4.2 Feasibility of seeding in practical applications

The overseeding experiment led to a substantial increase in ice number concentration, reaching values on the order of $10^6$ per mass (kg$^{-1}$) (Figure 10). Here, we perform a simple estimation of the required amount of seeding substances based on the

best case (seeding over 24 km × 24 km at 7.2 km height) to discuss the feasibility of our experiments. Although the primary objective of this study was to identify effective conditions for cloud seeding to mitigate the localization of heavy rainfall, such an estimation is expected to provide an opportunity for discussing the feasibility of cloud seeding in real-world settings.

Assuming that an increase in ice number concentration on the order of $10^6$ kg$^{-1}$ occurs uniformly throughout the seeding-applied location—represented by a box of 24 km × 24 km × 500 m (approximately the thickness of one vertical layer)—the

total number of generated ice particles within the seeding volume can be estimated as follows:

$$24000 \text{ (m)} \times 24000 \text{ (m)} \times 500 \text{ (m)} \times 0.58 \text{ (kg m}^{-3}) \times 10^6 \text{ (kg}^{-1}) = 1.8 \times 10^{17}$$

where air density at the height of 7.2 km is approximately 0.58 kg m$^{-3}$.

Here, assuming the use of dry ice as the seeding substance, and given that dry ice is known to generate an extremely large number of ice crystals—approximately $10^{13}$ particles per gram—the total mass of dry ice required can be estimated as 1.8 ×

$10^4$ (g) (i.e., 18 kg). This can be regarded as a rough estimate of the amount of dry ice required to produce the observed increase



in ice number concentration at a single time step. The total amount needed for a 3-hr seeding operation would depend on the frequency of seeding application necessary to maintain a consistent enhancement in ice number concentration throughout the period. For instance, assuming seeding is performed once every minute over a 3-hour period (i.e., 180 times), the total required amount of dry ice would be $18 \times 180 = 3{,}240$ kg. The feasibility of deploying such a large quantity of dry ice—considering the requirements for storage, transport, and in-flight distribution—must be carefully evaluated in conjunction with aircraft specifications and operational constraints. We limited our analysis to a rough estimation to obtain the order of magnitude of the required amount of seeding substance. This estimate is intended to provide a foundation for more comprehensive evaluations in future studies.

## 5 Concluding remarks

This study investigated the potential of cloud seeding to mitigate extreme rainfall localization (i.e., overseeding) associated with mesoscale convective systems in Japan. Main findings are summarized as follows:

- The cloud seeding experiment led to reduced rainfall within the heavy rainfall region and increased rainfall downstream, demonstrating the hypothesized dispersal mechanism of "overseeding".
- Seeding in the mid–upper troposphere (7.2–8.6 km), where air temperature ranged from −22°C to −12°C, resulted in the most pronounced changes in rainfall amount.
- Expanding the seeding area upstream of the heavy rainfall area had greater impact than increasing vertical thickness of the seeding.
- The most effective seeding configuration (24 km × 24 km area at 7.2 km) achieved an 11.5% decrease in area-averaged 3-hr accumulated rainfall and a 32% decrease as the maximum reduction in 3-hr accumulated rainfall over the heavy rainfall region.

This study includes some limitations. This study represented cloud seeding by simply modifying the ice nucleus concentration using the double-moment microphysics scheme, which reflects the seeded condition instantaneously within a specified spatial extent. To date, most cloud seeding efforts have primarily targeted drought mitigation, and the effectiveness of seeding in modifying rainfall patterns within mesoscale convective systems remains insufficiently demonstrated. In this context, this study focuses on assessing the feasibility of mitigating rainfall localization through simple adjustments to aerosol concentrations. However, such an implementation does not account for the advection/diffusion of seeding substances and their subsequent changes and interactions with hydrometeors. Thus, our findings may include the effects of unrealistic behavior of ice nucleus. To address this limitation, future studies should consider performing an overseeding experiment using a scheme that explicitly represents the transport, dispersion, and mixing of seeding agents within the atmosphere. Such a scheme would allow for a more realistic simulation of the spatial and temporal evolution of seeding materials, including their interaction with local thermodynamic and dynamic conditions. For instance, Xue et al. (2013a; 2013b) developed silver iodide cloud seeding parameterization scheme implemented in the WRF model. Guo et al. (2024) employed an effective approach to represent

seeding substance dispersion and interaction, which utilizes the WRF-Chem module coupled with an aerosol-aware microphysics scheme (Thompson and Eidhammer, 2014). Explicitly representing such process would help capture not only
the direct microphysical effects of cloud seeding but also the secondary responses associated with aerosol-cloud-precipitation interactions. Furthermore, the seeding location and its temporal movement could be designed to more realistically represent operational seeding practices, such as actual aircraft flight trajectories. Such design is closely related to the model configuration including model grid size. Integrating these advancements in both seeding methodology and implementation strategy would enhance the realism and applicability of numerical experiments.

This study focused on the MCS event in 2014 as a representative case of MCS events with line-shaped rain band in Japan (i.e., Senjo-Kousuitai) (Kato, 2020). However, to enhance the robustness of the findings, it is necessary to conduct similar experiments across a range of MCS events. Moreover, conducting similar experiments for different types of weather extremes, such as tropical cyclones, would be valuable for understanding the effectiveness of seeding across various weather systems. Such experiments are left for future work.

## Code availability

The code used to perform the seeding experiment and post-processes including visualization can be provided from the corresponding author upon a reasonable request.

## Data availability

The NCEP FNL GDAS is available at ([https://rda.ucar.edu/datasets/d083003/](https://rda.ucar.edu/datasets/d083003/)). The WRF simulation results can be provided
from the corresponding author upon a reasonable request.

## Author contributions

Y.H.: Conceptualization, Methodology, Software, Validation, Formal Analysis, Investigation, Data Curation, Visualization, Funding acquisition, Writing (original Draft); J.M.: Software, Validation, Formal Analysis, Investigation, Data Curation, Visualization; S.K.: Conceptualization, Methodology, Funding acquisition, Project administration, Writing (review and
editing); Y.S.: Methodology, Software; S.C.: Methodology, Writing (review and editing); A.H.: Methodology, Investigation, Writing (review and editing); K.Y.: Methodology, Investigation, Writing (review and editing); T.F.: Investigation, Visualization.

## Competing interests

The authors declare that they have no conflict of interest



## Acknowledgements

This work was supported by JST Moonshot R&D Program (JPMJMS2389-5-3). This work was supported by "Joint Usage/Research Center for Interdisciplinary Large-scale Information Infrastructures (JHPCN)" and "High Performance Computing Infrastructure (HPCI)" in Japan (Project ID: jh240013).

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
