# Peer review of "Numerical Experiments of Cloud Seeding for Mitigating Localization of Heavy Rainfall: A Case Study of Mesoscale Convective System in Japan"

_EGUsphere, 2025_

## Author Comment (AC1)

Comment:

The authors present a numerical analysis of the potential for cloud seeding to mitigate the rainfall from an extreme historical MCS event in Japan. The paper is interesting and well-written and I think is suitable for publication with moderate revisions that would help strengthen the novelty of the study and to highlight an addition (in my opinion, important) limitation and direction for future work. I also have a few minor grammar suggestions. I doubt I caught all the grammar mistakes, which anyway are so small and infrequent that they do not detract from the quality of the work.

Response:

We thank the reviewer for this positive and encouraging evaluation of our manuscript. We appreciate the reviewer's assessment that the study is interesting, well written, and suitable for publication. We are also grateful for the constructive suggestions aimed at strengthening the novelty of the work and for highlighting an important limitation and direction for future research.

In response, we have revised the manuscript to better clarify these aspects and to explicitly acknowledge the additional limitation and future research needs. We have also carefully addressed the minor grammatical suggestions provided by the reviewer and conducted an overall review of the manuscript to further improve clarity and readability.

**In our responses, the additional text introduced in the revised manuscript is indicated in red.**

Comment:

My concern regarding novelty stems from the fact that very little information is provided on numerical cloud seeding studies, particularly those aimed at rainfall mitigation and those based in Japan. This lack of information on previous work made it impossible for anyone who isn't themselves familiar with the literature to understand how much was new about the present study. That said, the introduction section is already lengthy. Therefore, I recommend creating a new section entitled "Background" or something like that, following the introduction, that can provide further details on methods and findings of a few key previous studies, which will help frame the current work. Some points now provided in the introduction may perhaps be moved into that background section as well.

Response:

Thank you for your useful comment. Following this suggestion, we added a new section entitled **"Background"** as Section 2. In this section, we review previous numerical studies of cloud seeding and overseeding, with particular emphasis on studies aimed at rainfall mitigation and those conducted in Japan, and summarize their key methodologies and findings. To improve the clarity and focus of the manuscript, several discussions on overseeding that were originally included in the Introduction have been moved to the new Background section. As a result, the Introduction has been streamlined to better highlight the motivation and objectives of the present study, while the Background section provides the necessary context to clarify the novelty and positioning of our work. The following shows the content of the new Background section. Section 2.1 was moved from the Introduction, and Section 2.2 was newly added to review previous studies and clarify the motivation for the present study.

**2 Background**

**2.1 Concept of overseeding**

The concept of overseeding was succinctly outlined in the textbooks of Mason (1971) and Rogers and Yau (1989) and nicely summarized in Durant et al. (2008), describing a scenario in which introducing an excessive quantity of ice nuclei leads to the formation of a large number of small ice crystals in convective clouds containing supercooled water droplets. Such seeding practice is categorized as glaciogenic seeding (Hashimoto et al., 2015). Under such conditions, the competition for available moisture within the cloud becomes intense, inhibiting the growth of individual ice crystals to sizes sufficient for precipitation. When the concentration of artificially generated ice crystals significantly exceeds natural levels, the rapid increase in the number of simultaneously growing precipitation particles can result in a substantial reduction in their growth rates due to moisture depletion. Furthermore, the freezing of supercooled water releases latent heat, which strengthens the updraft and thereby reduces the sedimentation velocity of precipitation particles. Consequently, these processes would lead to a decrease in the size and sedimentation velocity of precipitation particles at the location of the overseeding, which may temporarily suppress precipitation from the seeded cloud layer. Precipitation particles with reduced growth rates are likely advected downstream by upper-level wind and eventually fall as precipitation in the downstream region (i.e., redistributing rainfall over a broader area). Such a dispersal mechanism has the potential to mitigate the localization of intense precipitation. The aforementioned concept of overseeding has also been discussed in recent studies (Koloskov et al., 2010; Murakami, 2015; Korneev et al., 2022; Abshaev et al., 2022), leading to growing interest in its potential for disaster risk reduction.

**2.2 Numerical overseeding studies in Japan**

Although numerical investigations of cloud overseeding generally remain limited, a series of recent studies in Japan have suggested its potential to influence peak rainfall intensity during heavy rainfall events. Early studies using mesoscale atmospheric models showed that excessively increasing ice nucleus number concentrations within MCSs can reduce maximum rainfall intensity and the spatial extent of heavy rainfall (Suzuki et al., 2012; Onaka and Suzuki, 2014). In these studies, overseeding was typically represented by artificially multiplying the ice nucleus number concentration by large factors within cloud microphysics schemes.

Subsequent studies further examined the dependence of rainfall mitigation on cloud developmental stage and seeding strategy. Yokoyama et al. (2015) demonstrated that seeding during the early stage of cloud development was more effective than seeding during the mature stage, as early-stage seeding weakened vertical updrafts, leading to reductions in peak rainfall intensity. More recently, Nozaki et al. (2024) showed that the effectiveness of cloud seeding strongly depends on seeding location, timing, and environmental conditions, with both mitigation and enhancement of rainfall occurring under different configurations. In addition, studies exploring more targeted seeding strategies, such as pinpoint seeding, have suggested that focusing seeding on strong updraft regions may enhance mitigation efficiency (Yagi et al., 2017; Sano et al., 2024). These studies reported reductions in peak rainfall intensity on the order of 10–30% under favorable conditions and underscored the importance of seeding geometry and vertical placement.

Despite these advances, systematic comparisons of key seeding parameters, including seeding altitude, horizontal extent, and vertical thickness, remain limited, and their influence on the effectiveness of cloud seeding has not been fully clarified. Moreover, the physical mechanisms by which overseeding modifies microphysical growth pathways and interacts with mesoscale dynamics—particularly the downstream redistribution of precipitation in organized convective systems such as MCSs—remain insufficiently understood. These limitations motivate the present study, which aims to provide a more systematic and process-oriented assessment of rainfall responses to cloud overseeding.

*All the references were cited in the original manuscript.

Comment:

My other major concern is the lack of mention of what I see as the biggest limitation of the work and one of the biggest questions that needs to be answered if such approaches are ever to be operationalized. The authors examine the effects of cloud seeding directly into the storm area. In an operational context, this is akin to knowing exactly where the storm will take place. MCSs are notoriously difficult to forecast, particularly with respect to precise timing and location. Therefore, it seems unrealistic that such seeding could be accomplished in so localized a way. Instead, one would presumably need, with some multi-hour lead time, a prediction that a larger region is likely to develop an MCS somewhere within it, and seeding is done throughout the region. It is easy to imagine several things as a consequence of this: 1) the amount of AgI or other material needed could increase by orders of magnitude, and 2) the likelihood that some adverse outcome occurs due to rainfall being shifted spatially, rather than mitigated entirely, will increase, perhaps substantially. I suggest that the authors consider follow-up work that examines seeding, perhaps at lower concentrations, over larger areas, rather than just the storm location, to examine this effect. It may also be wise to perform ensemble simulations using some perturbation approach to better understand how stochastic the result may be.

Response:

Thank you for raising this important point regarding operational feasibility. We agree that prescribing seeding directly within the storm region implicitly assumes accurate knowledge of the timing and location of MCS development, which is challenging in practice. In the present study, we intentionally adopted a retrospective, mechanism-oriented design because our primary objective is to examine whether the hypothesized overseeding processes occur and to clarify their underlying physical mechanisms, rather than to assess real-time operational implementation. To achieve this objective, it is necessary to conduct experiments under well-understood conditions, such as known locations of deep convection and the presence of supercooled liquid water, where the proposed physical mechanisms can be robustly evaluated.

Nevertheless, we fully agree that careful consideration of operational implementation is essential for assessing the practical applicability of the proposed approach. Demonstrating practical feasibility, however, would require a comprehensive assessment of the current forecasting environment in Japan, including the accuracy of MCS predictions, observational constraints such as data availability and latency, and the ability to determine/flight appropriate seeding extent and timing under forecast uncertainty. Japan has a dense observation network, including ground precipitation radar, which may be useful to judge the seeding practice. Additional analysis design for assessing the practical feasibility should reflect such forecast environment accurately, while these are inherently case and

location specific. We believe that such additional analyses are beyond the scope of the present mechanism-focused study. We would like to explicitly address these issues as limitations and identify them as important topics for future research.

To incorporate the points discussed above into the manuscript, we introduced a new subsection entitled "4.3 Limitations and implications for operationalization" and added the following paragraph:

Second, consideration of practical feasibility from an operational perspective is lacking. The present study prescribes cloud seeding within regions where deep convection and supercooled liquid water are known to exist, based on retrospective simulations. This design was intentionally adopted to prioritize the examination of whether the hypothesized overseeding processes occur and to clarify their underlying physical mechanisms under well-understood conditions, rather than to assess real-time operational implementation. From an operational perspective, however, careful consideration of practical feasibility is essential. Demonstrating such feasibility would first require a comprehensive assessment of the current forecasting environment in Japan, including the accuracy of MCS predictions, observational constraints such as data availability and latency, and the capability to determine appropriate seeding extent and timing under forecast uncertainty. At the same time, designing analyses to assess operational feasibility must accurately reflect these forecasting and observational conditions and is inherently case- and location-specific. Because such comprehensive assessments extend beyond the scope of the present mechanism-focused study, these operational considerations are explicitly identified here as limitations and are left as important topics for future research.

The limitations originally discussed in Section 5 ("Concluding Remarks") were relocated to the newly introduced section, resulting in the removal of Section 5.

Minor comments:

Throughout: "downwind" is likely better than "downstream" for atmospheric work

Response:

Thank you for your suggestions.

Downstream is now replaced with downwind throughout the manuscript.

We replaced Figure 2(a) with the one below to clearly indicate the wind direction and clarify the downwind direction.

[Figure]

**Figure 2. (a) WRF-simulated 3-hr accumulated rainfall with the path for vertical cross sections (red line and points), horizontal extent of cloud seeding (white dashed square), heavy rainfall region (>100 mm in 3 hours; black outline), and 850 hPa wind vectors at 16:20 UTC on August 19, 2014.**

L17 and 18; delete "s" from "altitudes" and "areas"

Response:

Addressed.

L25: "… decrease as the maximum reduction…" is awkward wording

Response:

The sentence is revised as follows:

The most effective seeding configuration (24 km × 24 km area at 7.2 km) achieved an 11.5% decrease in area-averaged 3-hr accumulated rainfall and a maximum reduction of 32% in 3-hr accumulated rainfall over the heavy rainfall region.

L66 and 69: delete uses of "the" except at the start of line 69

Response:

Addressed.

L70: delete "to date" as it is unnecessary

Response:

Addressed.

L88: change "in" to "on"

Response:

Addressed.

L89: the 100 mm/h is unclear. Is that a peak rate over some short time interval?

Response:

We revised the sentence as follows to clarify the meaning:

[…] exceeding a peak hourly rainfall rate of 100 mm/h […]

L97: delete "s" from convections

Response:

Addressed.

L103: "under a future climate"

Response:

Addressed.

L104: countermeasure is one word

Response:

Revised as "countermeasure".

Eqn 1: Does beta have a physical meaning? If so, describe it. Also, what is a more typical value in WRF?

Response:

Thank you for your comment.

In the standard WRF implementation of the Morrison double-moment microphysics scheme, the ice nucleus concentration is diagnosed using the Meyers formulation without β, which represents background ice nuclei under natural atmospheric conditions. In this study, β is introduced as an artificial, unitless multiplier to represent an overseeding condition, in which the effective ice nuclei concentration in the mixed-phase region is deliberately increased far beyond typical background levels.

Physically, β does not represent a directly measurable environmental parameter; rather, it serves as a modeling control parameter that enables a strong enhancement of ice nuclei to mimic the effects of intense glaciogenic seeding. By adopting a very large value of β, we ensure a rapid and substantial increase in ice nucleus concentration, allowing us to isolate and examine the microphysical and dynamical response of convective clouds to extreme ice nuclei perturbations.

We have added the following sentences to Section 2.3 to clarify the explanation above:

In WRF v4.1.2, the number of ice nucleus concentration is determined based on the Meyer's formula within the Morrison 2-moment cloud microphysics scheme (Meyers et al., 1992).

$$n_c = \exp\{-2.80 + 0.262 \times (273.15 - T)\} \times \beta \qquad (1)$$

where $n_c$ is the number of ice nucleus concentration per kilogram, $T$ is air temperature in kelvin, and $\beta$ is a unit less multiplier which is introduced for seeding experiments. In the WRF model, the Meyer's formula triggers the freezing of cloud droplets when the following conditions are met: a cloud water mixing ratio greater than $10^{-14}$ kg kg$^{-1}$ and an air temperature below –4°C (i.e., deposition freezing). In the present study, β is introduced as an artificial, unitless multiplier to represent an idealized overseeding condition, in which the effective ice nuclei concentration in the mixed-phase region is intentionally increased by several orders of magnitude.

L195 and Table 2: use "design" rather than "flow"

Response:

Replaced "flow" with "design".

L195: change "was" to "is"

Response:

Addressed.

Figure 4: remind the reader what the thick black polygon indicates

Response:

We have added the following sentence in the caption of Figure 4:

The black outline denotes areas where the 3-hr accumulated rainfall exceeds 100 mm in the CTL run.

---

## Author Comment (AC2)

Comment:

This manuscript presents a clear numerical case study of cloud overseeding aimed at reducing the localization of extreme rainfall during the August 2014 Hiroshima event. The sensitivity tests of seeding height and seeding area are well organized, and the results provide useful insight into when the model is most responsive. I recommend minor revisions focused on adding clearer explanation and justification of real-world feasibility, uncertainty, and the practical implications of increased rainfall downstream, so that readers can interpret the findings appropriately.

Response:

We thank the reviewer for this positive and constructive evaluation of our manuscript. We appreciate the recognition of the clarity of the numerical case study, the organization of the sensitivity experiments, and the usefulness of the results in identifying conditions under which the model is most responsive.

In response to the reviewer's suggestions, we have revised our manuscript to improve the clarity of the discussion on real-world feasibility, uncertainty, and the practical implications of downstream rainfall increases, so that readers can more appropriately interpret the findings. We believe these revisions have strengthened the manuscript while remaining consistent with its mechanism-focused scope.

**In our responses, the additional text introduced in the revised manuscript is indicated in red.**

Major comments

1. Feasibility and operational realism: Please expand the feasibility discussion so readers can understand what real deployment could look like for the most effective configurations, which occur in the mid- to upper troposphere and require a large horizontal footprint. Clarify the assumed operational geometry (in cloud, cloud edge, or inflow) and the plausible delivery platforms at the target altitude. Please also discuss practical constraints near vigorous convection, including turbulence, icing, and flight safety, and how these constraints limit where seeding can be performed.

Response:

We thank the reviewer for this important comment. We agree that clearer explanation of real-world feasibility and operational realism is essential for the appropriate interpretation of the results.

In the revised manuscript, we have expanded the discussion while maintaining the original positioning of operational feasibility as a limitation, rather than attempting to resolve it within the scope of this mechanism-focused study.

First, in Section 2.3 (Experimental settings), we clarified the interpretation of the prescribed box-type seeding by adding the sentences below. We explicitly state that the fixed Eulerian enhancement of ice nuclei represents an idealized, best-case configuration equivalent to effective in-cloud delivery to the mixed-phase region, and should be regarded as an upper-bound sensitivity experiment rather than a direct operational scenario.

The imposed enhancement of ice nuclei within a fixed Eulerian box represents an idealized configuration equivalent to effective in-cloud delivery to the mixed-phase region. This setup is intended to identify favorable conditions and underlying mechanisms of overseeding, which should be regarded as an upper-bound sensitivity experiment rather than a realistic operational implementation.

Second, in Section 4.3 (Limitations and implications for operationalization), we have added a discussion clarifying that practical feasibility is constrained not only by forecasting and observational uncertainties but also by flight safety near vigorous convection. We now explicitly describe how strong turbulence, icing hazards associated with abundant supercooled liquid water, and lightning activity typically restrict crewed aircraft operations to the cloud edge or inflow region rather than the convective core, thereby limiting realizable seeding geometry and coverage relative to the idealized configuration used in the numerical experiments. We briefly note that future advances in alternative deployment platforms, such as unmanned aerial vehicles (UAVs), may offer opportunities to reduce human risk, while emphasizing that their practical viability remains highly uncertain due to current technical and operational limitations. We further clarify that evaluating the real-world feasibility of cloud seeding for heavy rainfall mitigation requires comprehensive assessments that extend beyond meteorological and microphysical perspectives, including engineering constraints, operational logistics, safety regulations, cost–benefit considerations, and social and institutional acceptability.

In addition to forecasting and observational uncertainties, practical feasibility is also constrained by flight safety near vigorous convection. Strong turbulence, icing hazards associated with abundant supercooled liquid water, and lightning activity typically restrict crewed aircraft operations to the cloud edge or inflow region rather than the convective core. While advances in alternative deployment platforms, such as unmanned aerial vehicles (UAVs), may offer new opportunities to reduce human risk and access hazardous regions that are difficult for crewed aircraft to operate in (Henneberger et al., 2023; Kazim et al., 2025), the practical viability of such approaches remains highly uncertain.

Ultimately, assessing the real-world feasibility of cloud seeding for heavy rainfall mitigation requires not only meteorological and microphysical considerations, but also comprehensive evaluations from multiple perspectives, including engineering constraints, operational logistics, safety regulations, cost–benefit trade-offs, and social and institutional acceptability. Because such comprehensive assessments extend beyond the scope of the present mechanism-focused study, these operational considerations are explicitly identified here as limitations and are left as important topics for future research.

Henneberger, J., Ramelli, F., Spirig, R., Omanovic, N., Miller, A. J., Fuchs, C., ... & Lohmann, U. (2023). Seeding of supercooled low stratus clouds with a UAV to study microphysical ice processes: an introduction to the CLOUDLAB project. *Bulletin of the American Meteorological Society*, *104*(11), E1962-E1979.

Kazim, M., Azzam, R., Burger, R., Wehbe, Y., Zweri, Y., Seneviratne, L., & Werghi, N. (2025). Unmanned aircraft systems for precipitation enhancement: Advancements, challenges, and future prospects. *Atmospheric Research*, 108333.

These additions are intended to provide readers with clearer operational context and uncertainty, while reinforcing the interpretation of the present results as upper-bound sensitivity experiments rather than immediately actionable operational guidance.

2. Targeting constraints in real experiments: Please describe limitations that arise from limited control over seeding location and timing in practice. In particular, discuss whether the reported optimal 24 km by 24 km case depends on perfect targeting of evolving upstream convective cells and how sensitive the conclusions might be if seeding must be restricted to smaller areas, shorter durations, or locations and timings offset from convective cores due to forecast and nowcast uncertainty.

Response:

We thank the reviewer for this insightful comment regarding targeting constraints in practical seeding operations.

The sensitivity experiments presented in this study were designed to explicitly examine how rainfall responses depend on the spatial extent and placement of seeding, particularly with respect to upstream convective cells. In Section 3.2, we systematically compare seeding areas of 6 km × 6 km, 12 km × 12 km, and 24 km × 24 km, and show that restricting seeding to smaller areas results in substantially weaker mitigation of localized heavy rainfall. This comparison indicates that the reported optimal 24 km × 24 km case

reflects conditions under which seeding successfully influences upstream convective development.

In addition, the analysis of divided seeding configurations in the Supplementary Material (Figures S7 and S8) demonstrates that seeding applied only to part of the upstream region is less effective than seeding over the full 24 km × 24 km area. These results highlight the sensitivity of the seeding impact to the ability to capture evolving upstream convective cells.

With respect to targeting accuracy in time and space, the experiments are conducted using retrospective simulations in which the evolution of the convective system is known a priori. We added discussion in Section 4.3 (Limitations and implications for operationalization), saying that such conditions differ fundamentally from real-time operations, where forecast and nowcast uncertainty would limit precise control over seeding location, duration, and timing. These constraints imply that the effectiveness achieved in the optimal 24 km × 24 km configuration should be interpreted as an upper-bound sensitivity result, and that reduced effectiveness can be expected if seeding must be applied over smaller areas, shorter durations, or locations offset from convective cores in practice.

Second, consideration of practical feasibility from an operational perspective is lacking. The present study prescribes cloud seeding within regions where deep convection and supercooled liquid water are known to exist, based on retrospective simulations. This design was intentionally adopted to prioritize the examination of whether the hypothesized overseeding processes occur and to clarify their underlying physical mechanisms under well-understood conditions, rather than to assess real-time operational implementation. From an operational perspective, however, careful consideration of practical feasibility is essential. Demonstrating such feasibility would first require a comprehensive assessment of the current forecasting environment in Japan, including the accuracy of MCS predictions, observational constraints such as data availability and latency, and the capability to determine appropriate seeding extent and timing under forecast uncertainty. At the same time, designing analyses to assess operational feasibility must accurately reflect these forecasting and observational conditions and is inherently case- and location-specific. […] Because such comprehensive assessments extend beyond the scope of the present mechanism-focused study, these operational considerations are explicitly identified here as limitations and are left as important topics for future research.

This interpretation is consistent with the study's emphasis on identifying favorable conditions and underlying mechanisms, rather than on prescribing operational seeding strategies.

3. Physical meaning of the seeding method: Please provide additional justification for representing overseeding by multiplying the ice nuclei concentration in the Meyers formulation by a very large factor within a fixed box. Readers will benefit from a direct statement of what magnitude of ice number enhancement is actually produced in the seeded volume during the seeding period (for example, typical and peak values). This would clarify how the chosen multiplier relates to a plausible range of effective ice nuclei increases and how strongly the results depend on this choice.

Response:

We thank the reviewer for this comment regarding the physical meaning and justification of the seeding representation.

In this study, the justification of the seeding method is based on the resulting ice number concentrations actually produced in the simulations, rather than solely on the choice of the artificial multiplier itself. As described in the caption and discussion of Figure 10, the seeding experiments produce an increase in ice number concentration on the order of $10^6$ $kg^{-1}$ at the seeding location during the active seeding period (e.g., at 16:30 UTC). This value represents the typical magnitude of enhancement achieved in the most effective cases.

To evaluate whether such ice number concentrations are physically plausible, we further relate them to known properties of intense glaciogenic seeding materials. Dry ice is known to generate an extremely large number of ice crystals through deposition freezing, on the order of $10^{13}$ $kg^{-1}$ per gram of dry ice (Fukuta et al., 1971; Murakami, 2015). Based on this established relationship, we estimated the amount of dry ice required to produce the simulated ice number enhancement, as presented in Section 4.2, resulting in an order-of-magnitude estimate of approximately 3,240 kg for a 3-hour seeding operation. This estimation provides the basis for our discussion of the practical realism and limitations of the seeding experiment. Therefore, the magnitude of ice number enhancement, its physical plausibility, and its implications for real-world feasibility are discussed explicitly in Section 4.2.

In contrast, Section 2.3 focuses on clarifying the modeling rationale of the seeding implementation. In response to the reviewer's comment, we revised Section 2.3 (shown below) to more clearly explain that applying a large artificial multiplier to the Meyers formulation within a fixed box is intended to represent an idealized overseeding condition, designed to isolate the microphysical and dynamical response of convective clouds to

extreme increases in ice nuclei, rather than to simulate the detailed transport and dispersion of seeding materials or a specific operational release rate.

Next, we performed cloud overseeding experiments by modifying the CTL run based on the concept of overseeding to investigate the potential of the overseeding for mitigating the heavy rainfall. We represented cloud overseeding in the numerical simulation by artificially increasing the number of ice nucleus concentration, following Suzuki et al. (2012), Yokoyama et al. (2015), and Nozaki et al. (2024). In WRF v4.1.2, the number of ice nucleus concentration is determined based on the Meyer's formula within the Morrison 2-moment cloud microphysics scheme (Meyers et al., 1992).

$$n_c = \exp\{-2.80 + 0.262 \times (273.15 - T)\} \times \beta \qquad (1)$$

where $n_c$ is the number of ice nucleus concentration per kilogram, $T$ is air temperature in kelvin, and $\beta$ is a unit less multiplier introduced in this study. In the WRF model, the Meyer's formula triggers the freezing of cloud droplets when the following conditions are met: a cloud water mixing ratio greater than $10^{-14}$ kg kg$^{-1}$ and an air temperature below –4°C (i.e., deposition freezing). We artificially increased the number of ice nucleus concentration by adopting a large unitless multiplier ($\beta=10^{11}$). This type of representation has been commonly adopted in previous numerical studies to examine the limiting microphysical and dynamical response of convective clouds to extreme glaciogenic seeding (e.g., Suzuki et al., 2012; Yokoyama et al., 2015; Nozaki et al., 2024). Such a large number of ice nuclei can be artificially generated using silver iodide (AgI) or dry ice (Fukuta et al., 1971).

Together, these clarifications connect the numerical implementation, the resulting ice number concentrations, and the subsequent feasibility discussion in a consistent framework.

4. Downstream impacts and net-hazard framing: Because the mechanism redistributes rainfall downstream, the results should be discussed in a net-hazard context. Please add a concise diagnostic or justification showing whether overall extremes are reduced rather than shifted. At minimum, report whether the domain-wide maximum 3-hour rainfall increases or decreases in the seeded runs and how the area exceeding key thresholds changes when computed from each seeded run, not only using the control-defined heavy-rainfall mask.

Response:

Thank you for this important comment. We agree that the results should be interpreted in a net-hazard context, given that the overseeding mechanism redistributes rainfall downstream.

To address this point, we examined rainfall changes over an extended analysis domain (131.8°E–132.8°E, 34°N–35°N), which encompasses not only the heavy-rainfall region defined in the control run but also the downstream area where rainfall increases were observed. This analysis is presented in Figures 8d–8f. In particular, Figure 8f shows the distribution of 3-hour rainfall differences over the extended area for the most effective seeding case (24 km × 24 km). The results indicate that negative rainfall differences dominate, with a minimum change of −45.43 mm and a maximum increase of +23.69 mm in 3-hr accumulation, suggesting that the overall extremes are reduced rather than simply shifted downstream.

These results indicate that the implemented seeding redistributes precipitation in a manner that mitigates the localization of extreme rainfall, without introducing new hazardous extreme accumulations in the downstream region at comparable magnitudes. While downstream rainfall increases do occur, their magnitude remains smaller than the reductions within the original heavy-rainfall region.

We note that we did not focus on domain-wide extrema over the full model domain, as such large-scale statistics are strongly influenced by atmospheric chaos and may obscure the localized response directly attributable to the seeding perturbation. Instead, we focused on a physically relevant regional scale aligned with the convective system of interest.

Based on these analyses, the manuscript concludes that the implemented seeding configuration reduces the hazard associated with localized extreme rainfall in a net-hazard sense, rather than merely displacing extremes downstream.

To clarify the point above, we added the following sentences to explain Figure 8:

[…] However, our evaluation over the extended analysis domain (131.8°E–132.8°E, 34°N–35°N) shows that the 24 km × 24 km seeding configuration induced greater rainfall increases further downstream (up to 23.7 mm; 310.5% in Figure 8f), compared to the 12 km × 12 km case resulting in a smaller maximum increase of 18.45 mm (241.8%; Figure 8e). Despite such increases downstream, reductions in extreme 3-hr rainfall remain dominant, indicating that the seeding primarily mitigates the localization of extremes rather than simply shifting comparable hazards downstream (Figure 8f).

5. Uncertainty and robustness: The manuscript notes strong nonlinearity and sensitivity to atmospheric chaos, but the results are interpreted mainly from single realizations. Please add a short justification for how robust you expect the sign and magnitude of the rainfall changes to be relative to internal model variability for this convective case. If additional ensemble simulations are not feasible, clearly state this limitation and adjust wording so conclusions are framed as conditional sensitivity results.

Response:

We appreciate your insightful comment. In response to this comment, we added a third limitation in Section 4.3 (shown below), clarifying that the present results are based on single realizations and are therefore subject to internal model variability associated with convective chaos. We now explicitly state that ensemble simulations would be required to assess robustness by averaging out chaotic variability and isolating the direct seeding signal, but that such simulations are left for future studies due to significant computational demand.

Third, the robustness of the simulated rainfall responses is subject to uncertainty arising from internal model variability and atmospheric chaos. The present analysis is based on single realizations for each seeding configuration, and the resulting rainfall changes reflect the response of a highly nonlinear convective system to a prescribed perturbation. While the physical consistency of the simulated microphysical responses supports the proposed overseeding mechanism, the magnitude and spatial pattern of rainfall changes may be influenced by internal variability inherent to convective-scale simulations. From this perspective, ensemble simulations would be essential for a more rigorous assessment of robustness. By considering multiple realizations with perturbed initial conditions, ensemble approaches can help reduce chaotic variability that is not directly related to seeding and thereby more effectively isolate the direct signal attributable to the seeding perturbation. Such analyses would allow a clearer quantification of the robustness of both the sign and magnitude of rainfall changes relative to internal variability.

Minor comments

6. Clarify the seeding time protocol in the model: whether the multiplier is applied continuously at each model time step from 16:00 to 19:00 UTC, or applied intermittently.

Response:

Thank you for pointing out this issue. We have added the following sentence in Section 2.3 (Experimental settings) to clarify the time step used in the seeding experiments:

In all overseeding experiments, the unitless multiplier in Meyer's formulation was applied at every model time step (every 3.33 seconds in d03) from 16:00 to 19:00 UTC on 19 August 2014.

7. For reproducibility, consider providing the specific code modification location where the multiplier is applied, as well as the key namelist settings used for these experiments.

Response:

Thank you for pointing out this issue. We have included the following sentence in Section 2.3 to clarify where we have made the modification to implement seeding practices:

The modification of Meyer's formulation was implemented within the WRF physics module *module_mp_morr_two_moment.F*.

Furthermore, we have added the important computational settings in Table 1 for clarifying our namelist settings:

| Physics parameterizations | |
|---|---|
| Cumulus convection (only d01) | Kain–Fritsch (Kain, 2004) |
| Cloud microphysics | Morrison 2-momnet (Morrison et al., 2009) |
| Shortwave radiation | RRTMG (Iacono et al., 2008) |
| Longwave radiation | RRTMG (Iacono et al., 2008) |
| Planetary Boundary Layer | MYNN 2.5 (Nakanishi and Niino., 2006; 2009) |
| Surface Layer | Revised MM5 (Jimenez et al., 2012) |
| Land surface processes | Noah-MP Land Surface Model (Niu et al., 2011; Yang et al., 2011) |
| Computational settings | |
| Time step [s] | 30 / 10 / 3.33 (d01 / d02 / d03) |
| Model integration period | 00:00 to 19:00 UTC on 19 August 2014 |

| Vertical layers | 70 layers, with model top at 50 hPa |
|---|---|
| SST update frequency | Every 6 hours |

8. Check the following typos:

- "Morrison 2-momnet" in Table 1 (should be "2-moment")

Response:

Revised as suggested.

- "upper raw, middle raw, and bottom raw" in the Figure 6 title (should be "row")

Response:

Revised as suggested.

- "folloing" in Table 2 (should be "following")

Response:

Revised as suggested.